# Frequency and diversity of carbapenemase-producing *Enterobacterales* recovered from untreated wastewater impacted by selective media containing cefotaxime and meropenem in Ohio, USA

Rael J. Too[1,2,3]*, George C. Gitao[2], Lilly C. Bebora[2], Dixie F. Mollenkopf[3], Samuel M. Kariuki[1], Thomas E. Wittum[3]

1 Kenya Medical Research Institute (KEMRI-Kenya), Nairobi, Kenya, 2 The University of Nairobi (UoN-Kenya), Nairobi, Kenya, 3 The Ohio State University (OSU-OH, USA), Columbus, OH, United States of America

* too.4@osu.edu

**Data Availability Statement:** All relevant data are within the paper.

## Abstract

As safe agents of last resort, carbapenems are reserved for the treatment of infections caused by multidrug-resistant organisms. The impact of β-lactam antibiotics, cefotaxime, and meropenem on the frequency and diversity of carbapenemase-producing organisms recovered from environmental samples has not been fully established. Therefore, this methodological study aimed at determining β-lactam drugs used in selective enrichment and their impact on the recovery of carbapenemase-producing *Enterobacterales* (CPE) from untreated wastewater. We used a longitudinal study design where 1L wastewater samples were collected weekly from wastewater treatment plant (WWTP) influent and quarterly from contributing sanitary sewers in Columbus, Ohio USA with 52 total samples collected. Aliquots of 500 mL were passed through membrane filters of decreasing pore sizes to enable all the water to pass through and capture bacteria. For each sample, the resulting filters were placed into two modified MacConkey (MAC) broths, one supplemented with 0.5 μg/mL of meropenem and 70 μg/mL of $ZnSO_4$ and the other supplemented with 2 μg/mL cefotaxime. The inoculated broth was then incubated at 37˚ C overnight, after which they were streaked onto two types of correspondingly-modified MAC agar plates supplemented with 0.5 μg/mL and 1.0 μg/mL of meropenem and 70 μg/mL of $ZnSO_4$ and incubated at 37˚C overnight. The isolates were identified based on morphological and biochemical characteristics. Then, up to four distinct colonies of each isolate's pure culture per sample were tested for carbapenemase production using the Carba-NP test. Matrix-assisted laser desorption/ionization-time of flight (MALDI-TOF) mass spectrometry (MS) MALDI-TOF MS was used to identify carbapenemase-producing organisms. In total 391 Carba-NP positive isolates were recovered from the 52 wastewater samples: 305 (78%) isolates had $bla_{KPC}$, 73 (19%) carried $bla_{NDM}$, and 14 (4%) harbored both $bla_{KPC}$ and $bla_{NDM}$ resistance genes. CPE genes of both $bla_{KPC}$ and $bla_{NDM}$ were recovered in both types of modified MAC broths, with 84 (21%) having a $bla_{KPC}$ gene, 22 (6%) carrying $bla_{NDM}$ and 9 (2%) harbored both a $bla_{KPC}$

**Funding:** This research was funded by the One Health Eastern Africa Training (OHEART) program at the Ohio State University, the global one health initiative (GOHi) through the National Institutes of Health (NIH) Fogarty international center (grant number TW008650), the University of Nairobi and KEMRI. The funders had no role in the study design, data collection, analysis, decision to publish, or preparation of the manuscript.

**Competing interests:** The authors have declared that no competing interests exist.

and $bla_{NDM}$ of isolates recovered from MAC medium incorporated with 0.5ug/mL meropenem and 70ug/mL $ZnSO_4$. The most prevalent isolates were *Klebsiella pneumoniae*, *Escherichia coli*, and *Citrobacter spp*.

## 1. Introduction

Carbapenemase-producing *Enterobacterales* (CPE) are rare, highly multidrug-resistant bacteria, often associated with hospitalized patients [1]. In the European Union, the burden of antibiotic-resistant infections is estimated to be equivalent to the estimated burden of influenza, TB, and HIV combined, with important contributors being beta-lactamase and carbapenemase-producing bacteria [2]. Most carbapenemase genes are encoded on plasmids and can be shared by different species of bacteria via horizontal gene transfer [3]. Hospital facilities have been implicated as the potential source of CPE in the environment. This is attributable to untreated or inadequately treated hospital influents on the environment [4, 5]. A previous study of hospital effluents and municipal wastewater treatment plants (WWTP) in urban Ireland showed that 36% of carbapenem non-susceptible isolates (n-23) were carbapenemase producers [6]. Thus CPEs can be disseminated from healthcare settings through wastewater flows to WWTP [7]. WWTPs are not designed to sterilize wastewater, but do lower quantities of both commensal and pathogenic microbes. They may, therefore, serve as reservoirs for the dissemination of bacteria of clinical concern, such as CPEs, into receiving surface waters [8].

*Klebsiella pneumoniae* carrying the carbapenemase gene, $bla_{KPC}$, is one of the most problematic; it is responsible for a large number of nosocomial infections, with a marked global endemicity and focal points reported in the USA, Israel, and Greece [9]. The gene belongs to the Amber class A KPC β-lactamase and is also carried by *Enterobacter* spp, *Escherichia coli*, and *Pseudomonas aeruginosa*, among others [10, 11]. Other threats related to carbapenemase spread, include the emergence of metallo-β- lactamases (MBLs) among enterobacterial species [12, 13]. Among the MBLs, which require zinc at the active site [14], the New Delhi metallo *β*-lactamase ($bla_{NDM}$) gene is widely disseminated globally. The subclass B1 metallo β-lactamase, $bla_{NDM}$, is highly transmissible within bacterial species; either carried by different incompatibility group plasmids (IncA/C, F, and L/M) or self-transmissible by conjugation [9, 15].

Detection of CPEs is important for the implementation of appropriate infection control measures [16]. Many studies have identified different approaches for the detection and/or recovery of CPEs. For example, Karthikeya *et al.* [13] used chromogenic selective culture media in both agar and broth, while Vrioni *et al.* [17] used ChromID CARBA which they found to be easily performed and a very accurate screening method. Mathys *et al.* [7] used MacConkey broth modified with 0.5 μg/mL of meropenem and 70 μg/mL of zinc sulfate and recovered 243 bacterial isolates which had reduced susceptibility to meropenem. Ninety of the recovered isolates (37%) exhibited the ability to hydrolyze carbapenem antimicrobials.

Currently, there is little information regarding the frequency and diversity of CPEs entering WWTPs and the potential impact of CPEs on the downstream ecosystem. Therefore, the purpose of this work was to determine the frequency and diversity of carbapenemase-producing *Enterobacterales* recovered from untreated wastewater in Ohio, USA, using MacConkey broth and agar incorporated with different concentrations of β-lactam antibiotics cefotaxime and meropenem. Hypothesizing that different β-lactam drugs used in the selective enrichment medium would impact the recovery of CPEs.

## 2. Materials and methods

### 2.1 Study area, sample collection, and processing

The study was carried out in Franklin County, Columbus, Ohio, USA between March and December 2019, where untreated wastewater samples from the Jackson Pike wastewater treatment plant (JP-WWTP) and pre-treated sanitary sewer mains were collected in 1L bottles for detection of the presence of CPEs. In total, 52 wastewater samples were collected with 22 samples collected from the JP- WWTP influent and 30 from 9 pre-treated sanitary sewer mains in Columbus, Ohio. Four of the pre-treated sewers namely Mill Run, Renner Road, and Southwest boulevard were received by Jackson pike WWTP while Sunbury Road 1 and 2 were not.

Wastewater influents were collected in polypropylene containers weekly from the WWTP and quarterly from the pre-treatment sites. Samples were recorded and aliquots of 500 mL were filtered using a Millipore vacuum filtration system apparatus, through filter papers of decreasing pore sizes (41μm, 20 μm, 10 μm, 1.2 μm, 0.8 μm, and finally 0.45 μm) (Thermo Fischer Scientific™ Nalgene™ Filter Membranes; Fischer Scientific, Hampton NH) to enable capture of all the bacteria present.

All the filters were aseptically placed into two types of modified MacConkey (MAC) broth (50 mL) in sterile Whirl packs: one supplemented with 0.5 μg/mL of meropenem and 70 μg/mL of $ZnSO_4$ and the second supplemented with 2 μg/mL cefotaxime. The inoculated broths were then thoroughly mixed [18] and incubated at 37˚ C overnight. After incubation, each broth was streaked onto two types of correspondingly-modified MacConkey (MAC) agar plates supplemented with either 0.5 μg/mL or 1.0 μg/mL meropenem and 70 μg/mL of $ZnSO_4$ and incubated at 37˚C overnight. Based on different morphological characteristics, up to four pure distinct lactose-positive colonies and one lactose-negative colony were picked from each MAC agar plate and inoculated onto Mueller Hinton agar (MHA) (BD BBL) supplemented with meropenem 0.5 μg/mL and 70 μg/mL $ZnSO_4$ and incubated at 37˚C overnight [18]. Then, for each sample, five isolated pure colonies were separately tested for carbapenemase production using the Carba-NP test. DNA extraction of Carba-NP test-positive isolates was then done and specific resistance genes were identified by conventional PCR.

### 2.2 Carba-NP test

Biochemical identification of carbapenemase production was done using the Carba-NP test directly from fresh bacterial cultures [19]. The method rapidly detects carbapenemase production using the *in vitro* hydrolysis of imipenem by bacterial lysates within approximately 2 hours after incubation at 37˚C by visualizing the resulting pH change using the color indicator, phenol red [19]. This method has been reported to be highly sensitive in the detection of *Klebsiella pneumonia* carbapenemase and metallo β-lactamase producers [20]. Suspect CPE colonies on MHA supplemented with 0.5μg/mL meropenem and 70 μg/mL $ZnSO_4$ were suspended in two 1.5mL micro-centrifuge tubes containing 100μl of 20mM Tris-HCL lysis buffer / SoluLyse™ solution (Bacterial Protein Extraction Reagent Buffer, pH 7.4) and then vortexed for 5 seconds. One micro-centrifuge tube was inoculated with 100 μL of aqueous indicator solution consisting of 0.05% phenol red with 0.1mmol/liter $ZnSO_4$, previously adjusted to pH 7.8, and 3mg/mL imipenem (Sigma) [19, 21]. The second tube was inoculated with 100 μL of the phenol red indicator solution without antibiotics, as a control. A red-to-yellow color change in the imipenem-enhanced tube after two hours of incubation indicated carbapenemase production.

### 2.3 DNA extraction

DNA extraction of Carba-NP test-positive isolates (those that hydrolyzed carbapenem), was accomplished by emulsifying colonies in 200 μL nuclease-free water, vortexing, and boiling at 100° C for 10 minutes. They were then centrifuged at 16,000 rpm for 3 minutes and the lysate supernates were stored at -20°C for molecular characterization of carbapenemase-producing resistance genes using conventional PCR.

## 2.4 Conventional polymerase chain reaction (PCR)

The screening was focused on the two most frequently reported carbapenemase-encoding genes in the US, $bla_{KPC,}$ and $bla_{NDM}$ [20], using primers published in other studies [20]. 3.0 μl of boiled, centrifuged lysate was used as a DNA template, while DNA from a standard control strain carrying $bla_{KPC}$ and $bla_{NDM-1}$ genes (ATCC®BAA-1705™ *Klebsiella pneumoniae* ART 2008133 [D-05, 1338] $bla_{KPC}$+/$bla_{NDM}$) were used as a positive control (Table 1).

Amplification reactions were performed [22, 23] at a reaction volume of 25 μL. Three μL of template DNA was added to a master mix containing PuReTaq™ Ready-To-Go™ PCR beads (Cytiva, Global Life Sciences Solutions, Marlborough, MA, USA), 1.5 μL of forward and reverse primers, and 19 μL of nuclease-free water to a final volume (Vf) of 25μL.

The thermocycling conditions for all the PCR amplifications of carbapenemase-encoding genes were as follows: 94°C for 5 minutes for initial denaturation and 30 cycles of 95°C for 30 seconds (denaturation), 52°C for 1 minute (all primers except for $bla_{KPC}$ at 58°C) and 72°C for 1 minute 30 seconds (extension), followed by 72°C for 10 minutes (final extension); performed in 0.2mL Eppendorf PCR tubes in a thermal cycler (MJ Research PTC-200). Amplified PCR products were separated and visualized by electrophoresis using a 1% agarose tris-acetate-EDTA gel containing 0.5 μg EtBr staining. To determine molecular weight, a 100–1,000 bp DNA ladder (Fermenters) was used.

### 2.5 Data analysis

Data were analyzed using a Microsoft Excel spreadsheet where descriptive statistics with binomial exact confidence intervals using frequencies and percentages of CPE prevalence were computed as the proportion of all the isolates. Associated resistance phenotypes and genotypes were computed as the proportion of wastewater with isolates and presented using frequencies and percentages.

## 3. Results

### 3.1 Frequency of recovered CPE from different sampling sites

In total, 391 carbapenemase-positives samples were recovered from the 52 wastewater samples collected. Of these, 305 (78%) isolates had $bla_{KPC}$ resistance gene, 73 (19%) carried a $bla_{NDM}$ resistance gene, and 14 (4%) of the isolates harbored both $bla_{KPC}$ and $bla_{NDM}$. Most isolates harboring $bla_{KPC}$ (n = 264, 68%) were identified from Jackson Pike WWTP while

**Table 1. Primers encoding for carbapenem-resistance.**

| Primer | Sequence (5'-3') | Target genes | TM (°C) | Size (bp) | References |
|--------|------------------|--------------|---------|-----------|------------|
| KPCyl-F | TGTCACTGTATCGCCGTCCTCAGTGCT CTA CAG AAA ACC | $bla_{KPC}$ | 58 | 750 | [22, 23] |
| KPCyl-R | | | | | |
| NDM1-F | CAG CGC AGC TTG TCG | $bla_{NDM}$ | 52 | 750 | [11] |
| NDM1-R | TCG CGA AGC TGA GCA | | | | |

**Table 2. Distribution of samples per study site.**

| Sampling sites | n | Carbapenemase-bearing isolates | | |
| --- | --- | --- | --- | --- |
| | | KPC | NDM | KPC and NDM |
| Jackson Pike WWTP Influent | 22 | 264 | 47 | 10 |
| Pre-treatment locations | | | | |
| Mill Run | 3 | 32 | 0 | 0 |
| Renner Rd | 3 | 4 | 13 | 4 |
| Sunbury 1 | 3 | 0 | 3 | 0 |
| Sunbury 2 | 3 | 5 | 0 | 0 |
| Southwest Blvd | 1 | 0 | 9 | 0 |
| Other 8 pre-treatment sites | 17 | 0 | 0 | 0 |
| Total Samples | 52 | 305 | 72 | 14 |

KPC- Klebsiella pneumonia carbapenemase, NDM- New Delhi metallo β-lactamase, n-number

sanitary sewer sites yielded 32 isolates from Mill Run (8%), 5 from Sunbury 2 (1%), and 4 isolates from Renner Rd (1%). Isolates harboring $bla_{NDM-1}$ were identified most from Jackson Pike WWTP (48, 12%), Renner Rd. (13, 3%), Southwest Blvd. (9, 2%) and Sunbury 1 (3, 1%). Thus, most isolates were from Jackson Pike WWTP; the majority carried $bla_{KPC}$ (Table 2).

### 3.2. Impact of cefotaxime vs meropenem selective enrichment treatment

Both of the CPE genes **$bla_{KPC}$** and **$bla_{NDM}$** were recovered in MAC broth modified with 2 μg/mL cefotaxime and 0.5 μg/mL meropenem, with 70% $ZnSO_4$, at 37°C overnight (Table 3).

### 3.3. Impact of meropenem concentration in selective agar

Of 391 isolates recovered, $bla_{KPC}$ and $bla_{NDM}$ genes were recovered on both 0.5μg/mL and 1.0μg/mL meropenem concentrations of MAC agar plates also supplemented with 70μg/mL $ZnSO_4$ (Table 4).

### 3.4. Distribution of isolates harboring both resistance genotypes and sequence types

Most of the isolates were recovered from meropenem enrichment broth modified with 0.5ug/mL meropenem and 70ug/mL $ZnSO_4$. The majority of carbapenemase-producing bacterial species included *K. pneumonia*, *E. coli*, and *Citrobacter spp*. (Table 5).

**Table 3. Recovery CPE using cefotaxime vs meropenem selective enrichment treatment.**

| Enrichment broth | KPC | | NDM | | KPC and NDM | |
| --- | --- | --- | --- | --- | --- | --- |
| | n | % of isolates | n | % of isolates | n | % of isolates |
| Meropenem/$ZnSO_4$ | 162 | 42% | 37 | 10% | 11 | 3% |
| Cefotaxime | 143 | 38% | 35 | 9% | 3 | 1% |
| Total | 305 | 79% | 72 | 19% | 14 | 4% |

KPC- Klebsiella pneumonia carbapenemase, NDM- New Delhi Metallo-beta-lactamase, n-number

**Table 4. Impact of different meropenem concentrations in selective agar.**

| | KPC | | NDM | | KPC and NDM | |
|---|---|---|---|---|---|---|
| **Selective MAC Agar** | **n** | **% of isolates** | **n** | **% of isolates** | **n** | **% of isolates** |
| **From meropenem broth:** | | | | | | |
| **0.5 μg /ml meropenem** | 84 | 21% | 22 | 6% | 9 | 2% |
| **1μg/ml meropenem** | 78 | 20% | 15 | 4% | 2 | 1% |
| From cefotaxime broth: | | | | | | |
| **0.5 μg /ml meropenem** | 76 | 19% | 15 | 4% | 2 | 1% |
| **1μg/ml meropenem** | 67 | 17% | 20 | 5% | 1 | 0% |

*KPC- Klebsiella pneumonia carbapenemase, NDM- New Delhi Metallo-beta-lactamase, n-number, μg/mL-microgram/milliliters.

## 4. Discussion

Hospital wastewater discharge to diverse aquatic settings, including WWTPs [15], has spread carbapenemase-producing *Enterobacterales* into the environment, posing a concern to public health. This study sought to investigate the effects of adding cefotaxime and meropenem β-lactam antibiotics to a selective medium as well as the frequency and diversity of CPEs collected from untreated wastewater influent in Columbus, Ohio. In this work, clinically significant CPEs were recovered from diverse residential pre-treated sanitary sewage sites that did not contain healthcare institutions as well as a large metropolitan WWTP utilizing various enrichment treatment methods. The CPEs of clinical concern were obtained from 5 of 9 (55.6%) pre-treatment locations and the influent of the WWTP. Jackson Pike WWTP receives waste from three of the five CPE-positive pre-treatment sites, but not from the Sunbury Road 1 and 2 sites. Suzuki et al. found 51 CPEs, which is similar to our work, with (47%) found in river water and (53%) being the majority from hospital waste [24]. Therefore, discharges into wastewater without prior treatment can significantly contribute to the spread of germs associated with hospitals, transferring bacteria like CPEs to the surface water from the realm of human healthcare [25]. Human healthcare ICUs, which have been characterized as factories for

**Table 5. Isolates harboring both resistance genotypes.**

| Isolate ID | Enrichment broth | Agar abx (meropenem) | Sequence types | Gene Expressed | Organism |
|---|---|---|---|---|---|
| JP WWTP wk 6 | meropenem* | 0.5μg/ml | KPC-2/NDM-5 | KPC | *K.pneumoniae* |
| JP WWTP wk 6 | cefotaxime | 1 μg/ml | KPC-2/NDM-5 | KPC | *E.coli* |
| JP WWTP wk 7 | meropenem | 0.5μg/ml | KPC-2/NDM-5 | NDM | *Citrobacter spp* |
| JP WWTP wk 7 | meropenem | 0.5μg/ml | KPC-2/NDM-5 | NDM | *Klebsiella. spp* |
| JP WWTP wk 8 | meropenem | 0.5μg/ml | KPC-2/NDM-5 | KPC | *K.pneumoniae* |
| JP WWTP wk 13 | meropenem | 1 μg/ml | KPC-2/NDM-5 | KPC | *K.pneumoniae* |
| JP WWTP wk 15 | meropenem | 0.5μg/ml | KPC-2/NDM-5 | KPC | *Citrobacter spp* |
| Renner Rd | meropenem | 0.5μg/ml | KPC-2/NDM-5 | NDM | *Citrobacter spp* |
| JP WWTP wk 19 | meropenem | 0.5μg/ml | KPC-2/NDM-5 | KPC | *Raoultella spp* |
| JP WWTP wk 20 | meropenem | 1 μg/ml | KPC-2/NDM-5 | NDM | *E.coli* |
| JP WWTP wk 21 | cefotaxime | 0.5μg/ml | KPC-2/NDM-5 | NDM | *Enterobacter spp* |
| JP WWTP wk 22 | cefotaxime | 0.5μg/ml | KPC-2/NDM-5 | NDM | *E.coli* |
| JP WWTP wk 22 | cefotaxime | 0.5μg/ml | KPC-2/NDM-5 | KPC | *K. pneumoniae* |

*Multiple species carrying both bla$_{KPC}$ and bla$_{NDM}$ genes were found at 2 locations. KPC- Klebsiella pneumonia carbapenemase, NDM- New Delhi Metallo-beta-lactamase, μg/mL-microgram/milliliters

producing, distributing, and amplifying antimicrobial resistance (AMR), are the source of CPEs [26]. As was to be expected, JP-WWTP influent samples made up the majority of samples containing clinically significant CPE isolates, with most isolates bearing the $bla_{KPC}$ gene and some carrying the $bla_{NDM}$ gene. Additionally, the $bla_{KPC}$ and $bla_{NDM}$ genes were present in the isolates from the pre-treatment sites. Despite being found in duplicate, the prevalence of (78%) $bla_{KPC}$ and (19%) $bla_{NDM}$ gene-harboring isolates is concerning and poses a serious threat to public health because they are resistant to last-resort medications. This is because they pose a threat to the public's health and can spread and invade healthcare settings, necessitating global cooperation and coordinated action. Carbapenem-resistant bacteria, including those found in this study, were recognized on a priority list of important pathogens causing diseases published by the World Health Organization (WHO) in 2017 [27]. The two most common sequence types in different *Enterobacterales species* were found in the $bla_{KPC-2}$ and $bla_{NDM-5}$ genes. The $bla_{KPC}$ gene was found in the current investigation in numerous residential pre-treatment sanitary sewer locations as well as WWTP influent. The frequency of the genotypes $bla_{KPC-2}$ and $bla_{NDM-5}$, which are carried by *K. pneumoniae*, *E. coli*, *Citrobacter*, and *Roultella spp*, is highlighted by the carbapenemase genes that were recovered and are shown in this paper. Notably, employing various enrichment treatment techniques from isolates that were found at a significant concentration. According to this study, MacConkey broth enriched with 0.5 μg/mL meropenem and 70 μg/mL ZnSO4 recovered CPEs at a rate of 42%, while MacConkey broth enriched with 2.0 μg/mL cefotaxime recovered CPEs at a rate of 38%. Nevertheless, meropenem concentrations of 0.5 μg/mL and 70 g/mL ZnSO4 selective agar from meropenem broth both demonstrated promising outcomes and can be used in future research. The first case of the internationally prevalent $bla_{KPC}$-2-bearing *K. pneumonia* was discovered in North Carolina in 1996 [28]. Clinical *Klebsiella michiganensis* [29] isolates also exhibited $bla_{KPC-2}$ detection and characterization, and following first reports, $bla_{KPC}$ rapidly disseminated throughout healthcare facilities and into the surrounding area via hospital effluents [7]. Similar to this study, the prevalence and diversity of these carbapenemase genes in WWTP have been shown by the identification of the $bla_{KPC}$ and $bla_{NDM}$ genes in raw influent and sanitary sewer flows. These sequence types were seen in isolates of *K. pneumoniae* [30] and *Enterobacter spp*. according to earlier research conducted elsewhere [20]. The data suggested that opportunistic pathogens as well as indigenous environmental bacteria, largely circulating in various sewer discharges in the US, may host the genes for $bla_{KPC-2}$ and $bla_{NDM-5}$. The growth and recovery of carbapenemase-producing bacteria from the sample have been improved with successive enrichment treatment choices of 0.5 g/mL meropenem with 70% ZnSO4 at 37˚C overnight in all selective media, both broth, and agar. The EFSA Panel on Biological Hazard showed in other experiments that pre-enrichment by incubation of samples in selective broth containing a carbapenem at a low concentration (e.g., meropenem 0.125mg/L) increased sensitivity [31]. Adams et al. [32] demonstrated that the growth of CPEs was facilitated by the presence of 0.5 μg/mL meropenem, 70 μg/mL ZnSO4, and 2.0 μg/mL cefotaxime when environmental and fecal samples were enriched and inoculated onto selective media. Although more CPEs were recovered in this study, their data is consistent with the results.

In contrast to our findings, Adler et al. [33] determined that the best approach for detecting CPE carriage in clinical isolates was MacConkey agar supplemented with imipenem at 1 μg/mL (McC+IMI), as opposed to a commercial selective medium. In contrast to what was discovered in this work, Pauly et al. [34] observed that MacConkey agar supplemented with 1.0 mg/L cefotaxime and 0.125 mg/L meropenem offered a greater sensitivity for the detection of CPE isolates. Vroni et al. [17] exhibited 85.2% sensitivity using direct plating of MacConkey agar plates supplemented with meropenem at 1 μg/mL, whereas in this work, a lower recovery rate of 20% and 4% for KPC and NDM-1, respectively, was found and this could be

attributable to increase in the concentration of meropenem. In this study, $bla_{KPC}$ and $bla_{NDM}$ were recovered in various concentrations of meropenem supplemented with 0.5 μg/mL or 1.0 μg/mL meropenem and 70 μg/mL ZnSO4 in sewage water from Jackson Pike WWTP. This study demonstrated the positive impact of meropenem concentration in selective MAC agar plates. Additionally, this study has confirmed that selective agar modified with 0.5 g/mL meropenem with 70 μg/mL ZnSo4 concentration incubated at 37˚C overnight improves the recovery of potential CPEs, even though the use of common selective culture methods does not appear to have an impact on the recovery of CPE from wastewater.

The study had a few drawbacks. Only carbapenemase isolates from WWTP influent and pre-treatment sewers, which represented all those discovered throughout the study period, were studied. Additionally, no background information has been gathered in hospital settings or other locations where high antimicrobial consumption contributing to wastewater was examined for this study. This information would be useful in comparing the findings of the current study. Although Mathys et al. highlighted the geospatial distribution of CPE from various sampling sites across the USA, additional carbapenemase-producing *Enterobacterales* will be researched in the future, including hospital discharges to compare and determine the sources and onsite pretreatment before discharge into municipal WWTP. Before now, WHO recommended onsite pretreatment of hospital effluents to remove potentially dangerous and toxic substances that could end up in the environment or aquatic ecosystem. Our study concentrated on influent and manhole sewers that did not contain hospital facilities because they are connected with threats to the public's health when wastewater that has not been properly treated or has not been treated is discharged. Although there are certain difficulties with onsite effluent treatment, WWTPs vary from one country to another [35]. It is a sign and a cause for concern that, despite WHO advise, there may be gaps that call for collective action. The recovery of potential CPEs within and between bacterial species is improved by using media modified with 0.5 μg/mL meropenem and 70 μg/mL ZnSO4 concentration and incubated at 37˚C overnight. This methodology study serves as a baseline for future studies and provides information on the ideal method to use.

## 5. Conclusion

This study has shown that there is a high prevalence of hospital-associated carbapenemase-producing bacteria being discharged into wastewater influents, indicating that WWTPs act as reservoirs for the dissemination of these carbapenemase-resistant organisms and their genes. Therefore, this is a critical public health concern and a global crisis that necessitates urgent and aggressive interventions. Carbapenemase-producing *Enterobacterales* carrying $bla_{KPC}$ and $bla_{NDM}$ were isolated in high frequency from WWTP influent and pre-treatment manhole sewers. Thus for areas where there is high antibiotic consumption, like hospitals and animal agriculture facilities, active and targeted surveillance of wastewater may be warranted to better understand the dynamics of the risk. The recovery of CPEs of multiple carbapenemase genes of clinical importance from the isolates, in this study, will provide data that will enable the relevant country authorities to formulate policies and intervention measures toward curbing the AMR situation.

## Acknowledgments

We acknowledge the contributions of Amy Albers for the technical support in sample processing, Dr. Gregory Ballash for manuscript review, Casey Hoffman for logistic support and Dr. Dubraska Diaz- Campos from clinical microbiology laboratory for provision of MALDI-TOF MS for organism identification.

We acknowledge the staff at GOHi international and Regional through One Health East Africa Research Training (OHEART) for capacity building and mentored training through NIH-D43 programme.

We appreciate the administration and staff of Jackson pike WWTP for facilitation of wastewater sample for this study. Special contributions by Director KEMRI, Dean, Department of Veterinary Preventive Medicine Ohio State University, and the Dean Department of Veterinary, Pathology & Microbiology and Parasitology, University of Nairobi for capacity building and various facilitation that aided the efforts of the Ph.D. fellowship throughout the study period.

## Author Contributions

**Conceptualization:** Rael J. Too, Thomas E. Wittum.

**Data curation:** George C. Gitao, Dixie F. Mollenkopf.

**Formal analysis:** Rael J. Too, Dixie F. Mollenkopf.

**Funding acquisition:** Thomas E. Wittum.

**Investigation:** Rael J. Too.

**Methodology:** Rael J. Too, Dixie F. Mollenkopf, Thomas E. Wittum.

**Project administration:** Dixie F. Mollenkopf.

**Resources:** Samuel M. Kariuki, Thomas E. Wittum.

**Supervision:** George C. Gitao, Lilly C. Bebora, Thomas E. Wittum.

**Validation:** Rael J. Too, Dixie F. Mollenkopf, Thomas E. Wittum.

**Visualization:** Rael J. Too.

**Writing – original draft:** Rael J. Too.

**Writing – review & editing:** Rael J. Too, George C. Gitao, Lilly C. Bebora, Dixie F. Mollenkopf, Samuel M. Kariuki, Thomas E. Wittum.

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
