## [Decision Letter · Decision Letter 0]

24 Oct 2022

PONE-D-22-27391Frequency and diversity of Carbapenemase-producing Enterobacteriaceae recovered from untreated wastewater impacted by β-lactam antibiotics Cefotaxime and Meropenem in Ohio, USAPLOS ONE Dear Dr. TOO,

Thank you for submitting your manuscript to PLOS ONE. After careful consideration, we feel that it has merit but does not fully meet PLOS ONE’s publication criteria as it currently stands. Therefore, we invite you to submit a revised version of the manuscript that addresses the points raised during the review process.

We look forward to receiving your revised manuscript.

Kind regards,

Mabel Kamweli Aworh, DVM, MPH, PhD. FCVSN

Academic Editor

PLOS ONE

Journal Requirements:

d) If you did not receive any funding for this study, please state: “The authors received no specific funding for this work.

5. We note that Figure 1 in your submission contain map images which may be copyrighted. All PLOS content is published under the Creative Commons Attribution License (CC BY 4.0), which means that the manuscript, images, and Supporting Information files will be freely available online, and any third party is permitted to access, download, copy, distribute, and use these materials in any way, even commercially, with proper attribution. For these reasons, we cannot publish previously copyrighted maps or satellite images created using proprietary data, such as Google software (Google Maps, Street View, and Earth). For more information, see our copyright guidelines: http://journals.plos.org/plosone/s/licenses-and-copyright.

6. Please upload a copy of Figure 2, to which you refer in your text on page 7. If the figure is no longer to be included as part of the submission please remove all reference to it within the text.

7. Please include a copy of Tables 2,4,6, and 8 which you refer to in your text on page 5 and 6.

Additional Editor Comments:

1. There were only five (5) tables presented in this manuscript. Please renumber the tables so that they correspond to the information already provided in the text. In the methods section, kindly reference "Table 1" in the text preceding the table as this is currently missing.

2. The discussion section needs to be rewritten. Please do not repeat the results in the discussion section, rather interpret the results in the light of previous studies and provide possible explanations for your findings.

3. Please highlight the limitations of this present study.

Reviewers' comments:

Reviewer's Responses to Questions

**Comments to the Author**

1. Is the manuscript technically sound, and do the data support the conclusions?

Reviewer #1: Yes

Reviewer #2: Yes

2. Has the statistical analysis been performed appropriately and rigorously? 

Reviewer #1: Yes

Reviewer #2: N/A

3. Have the authors made all data underlying the findings in their manuscript fully available?

Reviewer #1: Yes

Reviewer #2: Yes

4. Is the manuscript presented in an intelligible fashion and written in standard English?

Reviewer #1: Yes

Reviewer #2: Yes

5. Review Comments to the Author

Reviewer #1: Good publication, however, it seems like the author at least from the introduction fails to state the full objective of the study, on line 85 the author should consider revising and including that she/they were testing the concentration of the drugs in addition to the different types of drugs. On line 187, the author provides how different Meropenem concentrations faired but fails to provide how different concentrations of cefotaxime faired on the study. Were they just testing the concentration of Meropenem? This conflicts a finding on line 228 where they state the specific concentrations of Meropenem, and Cefotaxime showed good results without providing data on cefotaxime. Lastly, the author should provide more details on the DNA extraction method used

Reviewer #2: This is a very important paper as the AMR threat to global health persists. The more curated data we have to combat this threat from all fronts, the more efficient and fine-tuned our efforts would be.

The value of this particular work might be improved by providing background data on antimicrobial consumption in the hospitals/areas contributing to the waste water tested in the study. Such data can then be compared to the results of the present study. The authors stated "Thus for areas where there is high antibiotic consumption, like hospitals and animal agriculture facilities, active and targeted surveillance of wastewater should be carried out, to establish the dynamics of the situation". Data that actually shows the area studied actually has high antibiotic consumption will be helpful.

The authors may wish to know that the table labeling in the manuscript is overtly confusing. The genesis of this confusion remains unknown to the reviewer and may not enhance readers' experiences. Are some tables missing? The sic regular pattern by which this error is also bewildering. Even numbered tables (2, 4, 6 and 8) referred to the text are missing in the downloaded manuscript. Odd numbered tables not referred to in the text are available.

6. PLOS authors have the option to publish the peer review history of their article (what does this mean?). If published, this will include your full peer review and any attached files.

Reviewer #1: No

Reviewer #2: No

---

## [Author Response · Author response to Decision Letter 0]

24 Dec 2022

I hereby appreciate all the PLOS ONE Secretariat and the reviewers; one (1) and two (2) for the good review comments raised and suggestions to improve this manuscript. They were very helpful. Thank you.

The comments raised were addressed as follows;

Note: Corresponding email; changed to too.4@osu.edu

1. Plos -One style requirements- All the files are named accordingly.

2. PLOS questionnaire on Inclusivity – included as an attachment.

3. Financial disclosure;

a. I am a Ph.D. student at the University of Nairobi, OHEART – GOHi Ph.D. fellow at the Ohio State University (OSU) through the National Institutes of Health (NIH) Fogarty international center (grant number TW008650) This was a mentored Training through Ph.D. fellowship exchange program involving the three institutes (OSU, KEMRI and the University of Nairobi).

b. The funders had no role in study design, data collection, and analysis, decision to publish, or preparation of the manuscript

c. 1st Author student stipend (National Institutes of Health (NIH) Fogarty international center (grant number TW008650)

d. The authors received no specific funding for this work

4. Financial information: 

a. Rael Too (1st and corresponding author) 

b. grant number TW008650 (NIH D43 Fogarty PhD Fellow GOHi-OHEART-OSU)

c. Student stipend

5. Data Availability statement;

• The original contribution presented in this study is included in the study manuscript, further inquiries can be directed to the corresponding author.

6. Figure 1&2. No figures, (All deleted)

7. There is no figure 2, and any references to it are deleted and revised 

8. Tables -The Tables are corrected and captioned correctly see (Tables 1, 2, 3, 4, 5) 

9. Reference list 

The reference list (font changed -Times Roman 12), duplicate references edited additional references added and the list updated;

1. Azuma T, Otomo K, Kunitou M, Shimizu M, Hosomaru K, Mikata S, et al. Environmental fate of pharmaceutical compounds and antimicrobial-resistant bacteria in hospital effluents, and contributions to pollutant loads in the surface waters in Japan. Sci Total Environ. 2019 Mar;657:476–84.

2. Proia L, Anzil A, Borrego C, Farrè M, Llorca M, Sanchis J, et al. Occurrence and persistence of carbapenemases genes in hospital and wastewater treatment plants and propagation in the receiving river. J Hazard Mater. 2018 Sep;358:33–43.

3. Cahill N, O’Connor L, Mahon B, Varley Á, McGrath E, Ryan P, et al. Hospital effluent: A reservoir for carbapenemase-producing Enterobacterales? Sci Total Environ. 2019 Jul;672:618–24.

4. Suzuki Y, Nazareno PJ, Nakano R, Mondoy M, Nakano A, Bugayong MP, et al. Environmental Presence and Genetic Characteristics of Carbapenemase-Producing Enterobacteriaceae from Hospital Sewage and River Water in the Philippines. Elkins CA, editor. Appl Environ Microbiol. 2020 Jan 7;86(2):e01906-19.

5. Pauwels B, Verstraete W. The treatment of hospital wastewater: an appraisal. J Water Health. 2006 Dec 1;4(4):405–16

6. Cornaglia G, Akova M, Amicosante G, Cantón R, Cauda R, Docquier JD, et al. Metallo-β-lactamases as emerging resistance determinants in Gram-negative pathogens: open issues. Int J Antimicrob Agents. 2007 Apr;29(4):380–8. 

Additional Editor comments 

1. Tables 

There were only five tables presented in the manuscript and all are renumbered and now correspond to the information in the text. “Table 1” in the text preceding included (page 4 line-150)

2. Discussion section - rewritten 

3. Limitations of this present study included (page 9 lines 281-296)

4. Review comments to the author 

A. Reviewer #1: the full objective of the study,

i. “Testing the concentration of the drugs in addition to different types of drugs”

Only two β-lactam antibiotics were being tested (meropenem and cefotaxime)

ii. ‘How did Meropenem concentrations fair but fail to provide different (Line 243-249) concentrations of cefotaxime, were they testing meropenem concentration?’ Yes, we tested Meropenem concentration at 0.5 μg/mL and 1.0 μg/mL meropenem modified with 70µg/mL ZnSO4 to check what concentration was ideal to recover more CPEs.

iii. Line 228 specific concentrations of Meropenem, and cefotaxime showed good results without providing data on cefotaxime. Cefotaxime had low recovery rate compared to Meropenem in MacConkey broth (Line 243-249)

iv. Details on DNA extraction method included (page 4, lines 139-144)

B. Reviewer #2: AMR threats

i. Background data on AMR consumption in Hospitals/areas contributing to the wastewater tested in the study. (Done- Introduction section lines 61-65)

ii. Data that shows the area studied has high antibiotic consumption will be helpful. (Included lines 216-222)

iii. Table labeling in the manuscript Addressed

NB: Additional comments raised in recent submissions R1 and edits from the Supervisor

 About the mismatch in financial disclosure and financial information, this has been addressed as follows;

Financial disclosure;

a) I am a Ph.D. student at the University of Nairobi, OHEART – GOHi Ph.D. fellow at the Ohio State University (OSU) through the National Institutes of Health (NIH) Fogarty international center (grant number TW008650) This was a mentored Training through Ph.D. fellowship exchange program involving the three institutes (OSU, KEMRI and the University of Nairobi).

b) The funders had no role in study design, data collection, and analysis, decision to publish, or preparation of the manuscript

c) 1st Author student stipend (National Institutes of Health (NIH) Fogarty international center (grant number TW008650)

d) The authors received no specific funding for this work

Financial information: 

a. Rael Too (1st and corresponding author)

b. grant number TW008650

c. Student stipend

Other comments raised by the last author and have been addressed includes; 

a. Title edited to read; “Frequency and diversity of Carbapenemase-producing Enterobacterales recovered from untreated wastewater impacted by selective media containing cefotaxime and meropenem in Ohio, USA.”

b. replaced Enterobacteriaceae- Enterobacterales through out the manuscript.

c. Minor typos in throughout the Manuscript (Final Marked up copy with track changes)

---

## [Decision Letter · Decision Letter 1]

8 Jan 2023

PONE-D-22-27391R1Frequency and diversity of Carbapenemase-producing Enterobacterales recovered from untreated wastewater impacted by selective media containing cefotaxime and meropenem in Ohio, USAPLOS ONE

Dear Dr. TOO,

Thank you for submitting your manuscript to PLOS ONE. After careful consideration, we feel that it has merit but does not fully meet PLOS ONE’s publication criteria as it currently stands. Therefore, we invite you to submit a revised version of the manuscript that addresses the points raised during the review process.

We look forward to receiving your revised manuscript.

Kind regards,

Mabel Kamweli Aworh, DVM, MPH, PhD. FCVSN

Academic Editor

PLOS ONE

Journal Requirements:

Additional Editor Comments:

1. In discussing your results, please do not repeat the values as these have already been presented in the results section, for example you do not need to mention Tables in your discussion section rather provide the interpretation of your results and possible explanations for your findings while comparing your results with reports of other published works. The discussion section needs to be revised please.

Reviewers' comments:

Reviewer's Responses to Questions

**Comments to the Author**

1. If the authors have adequately addressed your comments raised in a previous round of review and you feel that this manuscript is now acceptable for publication, you may indicate that here to bypass the “Comments to the Author” section, enter your conflict of interest statement in the “Confidential to Editor” section, and submit your "Accept" recommendation.

Reviewer #1: All comments have been addressed

Reviewer #2: (No Response)

2. Is the manuscript technically sound, and do the data support the conclusions?

Reviewer #1: Yes

Reviewer #2: Partly

3. Has the statistical analysis been performed appropriately and rigorously? 

Reviewer #1: Yes

Reviewer #2: N/A

4. Have the authors made all data underlying the findings in their manuscript fully available?

Reviewer #1: Yes

Reviewer #2: Yes

5. Is the manuscript presented in an intelligible fashion and written in standard English?

Reviewer #1: Yes

Reviewer #2: Yes

6. Review Comments to the Author

Reviewer #1: No additional comments. The author has responded to all the comments raised adequately and as such, I propose that the document is pulished

Reviewer #2: The authors have rectified the issues around the tables as earlier raised.

However, the new references added by the authors are general and did not seem to reflect the specific area where their research was carried out. The original request was ...."The value of this particular work might be improved by providing background data on antimicrobial consumption in the hospitals/areas contributing to the waste water tested in the study. Such data can then be compared to the results of the present study. .... Data that actually shows the area studied actually has high antibiotic consumption will be helpful." Are previous studies on AMR and antibiotic consumption lacking for that area?

7. PLOS authors have the option to publish the peer review history of their article (what does this mean?). If published, this will include your full peer review and any attached files.

Reviewer #1: No

Reviewer #2: No

---

## [Author Response · Author response to Decision Letter 1]

14 Jan 2023

Response to Reviewers Manuscript R2.

Greetings! I hereby appreciate all of you the PLOS ONE Secretariat and the reviewers; one (1) and two (2) for the good review comments raised to improve this paper. They were very helpful. Thank you.

1. Laboratory protocols- N/A 

2. Discussion - Done

i. Repeated values were deleted and others used percentages to discuss.

ii. Mentioned tables deleted 

iii. The discussion section is revised 

3. Reference list 

The reference list is complete and correct

Additional Editor comments 

1. Review comments 

i. Reviewer #2: 

i. There was no background data collected on AMR consumption in hospitals/areas contributing to the wastewater tested in the study. 

ii. This is one of the limitations where by background data is missing or would have been collected and compare the results to ascertain the source (Line 286-290). However, this is a research gap and is part of the ongoing study in Kenya. 

iii. Discussion section revised.

N/B: The findings from this study informed on the ideal method to use in implementing the current study in Kenya.

---

## [Decision Letter · Decision Letter 2]

6 Feb 2023

Frequency and diversity of Carbapenemase-producing Enterobacterales recovered from untreated wastewater impacted by selective media containing cefotaxime and meropenem in Ohio, USA

PONE-D-22-27391R2

Dear Dr. TOO,

We’re pleased to inform you that your manuscript has been judged scientifically suitable for publication and will be formally accepted for publication once it meets all outstanding technical requirements.

Kind regards,

Mabel Kamweli Aworh, DVM, MPH, PhD. FCVSN

Academic Editor

PLOS ONE

Additional Editor Comments (optional):

Reviewers' comments:

Reviewer's Responses to Questions

**Comments to the Author**

1. If the authors have adequately addressed your comments raised in a previous round of review and you feel that this manuscript is now acceptable for publication, you may indicate that here to bypass the “Comments to the Author” section, enter your conflict of interest statement in the “Confidential to Editor” section, and submit your "Accept" recommendation.

Reviewer #1: All comments have been addressed

Reviewer #2: All comments have been addressed

2. Is the manuscript technically sound, and do the data support the conclusions?

Reviewer #1: Yes

Reviewer #2: Yes

3. Has the statistical analysis been performed appropriately and rigorously? 

Reviewer #1: Yes

Reviewer #2: N/A

4. Have the authors made all data underlying the findings in their manuscript fully available?

Reviewer #1: Yes

Reviewer #2: Yes

5. Is the manuscript presented in an intelligible fashion and written in standard English?

Reviewer #1: Yes

Reviewer #2: Yes

6. Review Comments to the Author

Reviewer #1: No additional comments. The author has responded to all the comments raised adequately and as such, I propose that the document is pulished

Reviewer #2: The authors of the manuscript have addressed all the concerns earlier raised. The manuscript is now acceptable

7. PLOS authors have the option to publish the peer review history of their article (what does this mean?). If published, this will include your full peer review and any attached files.

Reviewer #1: No

Reviewer #2: No

---

## [Editor Report · Acceptance letter]

10 Feb 2023

PONE-D-22-27391R2 

Frequency and diversity of Carbapenemase-producing *Enterobacterales* recovered from untreated wastewater impacted by selective media containing cefotaxime and meropenem in Ohio, USA 

Dear Dr. Too:

I'm pleased to inform you that your manuscript has been deemed suitable for publication in PLOS ONE. Congratulations! Your manuscript is now with our production department. 

Kind regards, 

on behalf of

Dr. Mabel Kamweli Aworh 

Academic Editor

PLOS ONE